# Estudio sobre el ajuste óptimo de pesos en SPODEs para balancear *Accuracy* y *Fairness* en el clasificador probabilístico AODE

**M. Julia Flores**∗
Departmento de Sistemas Informáticos - I3A
Universidad de Castilla - La Mancha
`Julia.Flores@uclm.es`

**José A. Gámez**
Departmento de Sistemas Informáticos - I3A
Universidad de Castilla - La Mancha
`Jose.Gamez@uclm.es`

## Abstract

Este proyecto propone un enfoque basado en optimización multiobjetivo y algoritmos evolutivos para parametrizar el clasificador Average One-Dependence Estimators (AODE), mediante un sistema de pesado de los diferentes SPODEs. El estudio está centrado en el impacto que este pesado produce respecto a la mejora en cuanto a medidas de fairness (equidad) para este clasificador, sin menoscabar en exceso su precisión. Se utilizará Naïve Bayes como baseline para comparar los resultados. La optimización se llevará a cabo mediante técnicas evolutivas como NSGA-II o MOEA/D, buscando configuraciones óptimas que reduzcan sesgos en conjuntos de datos con atributos sensibles. Se evaluará el impacto del ajuste de pesos en varios datasets, utilizando métricas estándar de clasificación y equidad. Finalmente, se analizarán los trade-offs entre precisión y fairness a través de la exploración de fronteras de Pareto, validando la viabilidad del enfoque propuesto.

## 1. Motivación

En los últimos años, el uso de modelos de aprendizaje automático en la toma de decisiones ha generado preocupación debido a los sesgos que pueden derivarse de los datos y los algoritmos utilizados [12]. Los modelos de clasificación automática juegan un papel crucial en la toma de decisiones en ámbitos como la sanidad, el crédito financiero y la justicia penal. En particular, los clasificadores bayesianos, aunque eficientes y robustos, pueden mostrar disparidades en la clasificación cuando se aplican a conjuntos de datos con atributos sensibles como género, etnicidad o edad. Esto ha evidenciado problemas relacionados, ya que se puede derivar en decisiones discriminatorias. Por ello, la comunidad científica y la sociedad demanda el desarrollo de modelos más justos, capaces de equilibrar precisión y equidad.

Dentro de la familia de clasificadores bayesianos, el Average One-Dependence Estimators (AODE) [15] ha destacado en rendimiento frente al clásico Naïve Bayes (NB) [11] y otros representantes de la familia semi-NB que también relajan la independencia condicional entre atributos [1]. Inicialmente, en el clasificador AODE, todos los SPODES aportan de manera equitativa. Sin embargo, podría aplicarse una técnica de ponderado donde se pudiera variar la importancia de los diferentes Super Parent One-Dependence Estimators (SPODEs) a la hora de realizar la clasificación. Esta estrategia de pesado va más alla de ponderar la importancia que cada variable tiene en el modelo final, ya que lo que se pondera es la importancia de un sub-modelo que incluye a todas las variables y modela explícitamente relaciones bivariadas entre algunas de ellas dada la clase. Ajustar estos pesos adecuadamente puede ser clave para mejorar tanto la calidad de la clasificación como cuán 'justa' esta es.

---

∗Julia.Flores@uclm.es

Preprint. Under review.

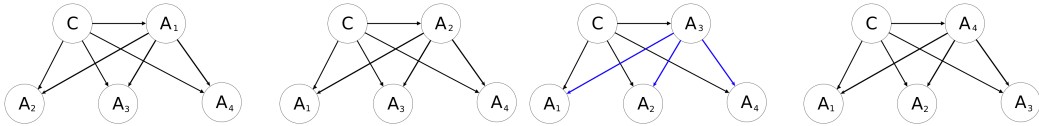

Figura 1: Figura donde se presentan los 4 SPODEs para una AODE con cuatro variables predictoras o atributos $A_i$.

Aunque existen variantes de AODE con pesos [9], hasta nuestro conocimiento nunca se han optimizado para propósitos de equidad. Este trabajo busca, por tanto, explorar cómo algoritmos evolutivos y optimización multiobjetivo pueden utilizarse para encontrar la mejor parametrización de pesos en AODE, maximizando métricas de rendimiento sin comprometer medidas de fairness.

## 2. Hipótesis de trabajo principal

En la literatura podemos encontrar distintos enfoques para abordar aspectos de equidad en el clasificador NB, como p.e. los centrados en garantizar la independencia de la clasificación respecto al valor de un atributo sensible [4], el uso de más de una variable sensible [3] o en el descubrimiento y eliminación de patrones discriminativos [6]. Sin embargo, no hemos encontrado ningún estudio centrado en el clasificador AODE [15], uno de los denominados clasificadores semi-NB más eficientes y eficaces.

AODE es un clasificador de los llamados generativos, puesto que en su aprendizaje trata de modelar la distribución de probabilidad conjunta $P(C, \mathbf{A})$ en lugar de $P(C|\mathbf{A})$. AODE trata a todas las variables por igual, sin considerar la existencia de variables sensibles. En este trabajo partimos de la asunción de que existe la posibilidad de mejorar simultáneamente la precisión y la equidad en clasificadores AODE mediante la optimización de los pesos asignados a cada SPODE, o al menos mejorar en equidad sin perjudicar en exceso la precisión. Puesto que se dispone de algoritmos evolutivos dirigidos a la optimización multiobjetivo [2], podríamos aplicarlo a nuestro tema de estudio, adaptándolo también a Naive Bayes, para compararlo con un clasificador bayesiano más sencillo. Diferentes estrategias de optimización pueden llevar a distintos trade-offs entre precisión y fairness, lo que sugiere la necesidad de analizar el frente de Pareto para seleccionar la mejor configuración según el caso de uso.

Consideramos que es un ámbito de estudio potencialmente interesante, y que es diferente al pesado de variables [14] o de instancias [13], puesto que en el AODE se combinan SPODEs (ver Figura 1), resultando en una agregación de múltiples clasificadores con restricciones estructurales para facilitar el aprendizaje y limitar su complejidad.

Para determinar la clasificación de una instancia se emplea la siguiente expresión:

$$\hat{c} = \arg\max_C \sum_{i=1}^{n} P(C)P(A_i \mid C) \prod_{\substack{j=1 \\ j \neq i}}^{n} P(A_j \mid A_i, C)$$

donde:

- $P(C)$ es la probabilidad a priori de la clase.
- $P(A_i \mid C)$ es la probabilidad del superpadre dado la clase, para cada uno de los SPODEs.
- $P(A_j \mid A_i, C)$ es la probabilidad condicional de los otros atributos dado el superpadre y la clase.

El enfoque que proponemos, buscará integrar un vector de pesos optimizado: $w_1, w_2, \ldots w_n$ tal que la clasificación sea:

$$\hat{c} = \arg\max_C \frac{1}{\sum_{k=1}^{n} w_k} \times \sum_{i=1}^{n} w_i \cdot P(C)P(A_i \mid C) \prod_{\substack{j=1 \\ j \neq i}}^{n} P(A_j \mid A_i, C)$$

donde en definitiva cada SPODE será dotado de una importancia. Pretendemos usar pesos numéricos, y en principio los $w_i$ estarían en el intervalo [0,1].

## 3. Objetivos

El principal propósito es diseñar e implementar un método basado en optimización multiobjetivo con algoritmos evolutivos para ajustar los pesos en un clasificador AODE, con el fin de mejorar tanto la precisión como la equidad en la clasificación. Para ello, más específicamente, pretendemos:

1. Implementar un esquema de optimización de pesos para AODE que permita mejorar métricas de fairness sin comprometer significativamente la precisión.

2. Comparar el rendimiento del AODE ajustado con el AODE estándar y el Naïve Bayes como baseline.

3. Evaluar el impacto del ajuste de pesos en diferentes datasets con atributos sensibles.

4. Analizar los trade-offs entre precisión y fairness mediante la exploración de fronteras de Pareto. Para ello consideraremos distintas medidas de equidad de entre las disponibles en la literatura:

   - *Paridad Estadística:* La probabilidad de una predicción positiva, dada la pertenencia a un grupo, debe ser igual para todos los grupos.
   - *Impacto Dispar:* La media del cociente de predicciones positivas entre cada par de grupos debe ser 1 o mayor que un porcentaje $p\%$ determinado.
   - *Equidad Diferencial:* Aplicación de la equidad grupal a grupos definidos por múltiples atributos sensibles superpuestos.
   - *Equidad Individual:* La diferencia en la probabilidad de los resultados entre dos individuos no debe ser mayor que la distancia de similitud entre ellos.
   - *Equidad Causal:* Uso de modelado causal para determinar el efecto de los atributos sensibles en las predicciones.

## 4. Metodología

Este es el procedimiento que planeamos seguir:

1. Selección de datasets.- Se trabajará con datasets de clasificación que contengan variables sensibles, como COMPAS, Adult Income y German Credit, entre otros. [10]. Se discretizarán los atributos numéricos para su uso en clasificadores bayesianos. Implementación de los modelos

2. Implementación de modelos.- Naïve Bayes se usará como baseline, proporcionando un punto de comparación sin ajuste de pesos. AODE con ajuste de pesos será el modelo principal, donde los pesos de los SPODEs se optimizarán mediante algoritmos evolutivos.

3. Optimización de pesos.- Se utilizarán algoritmos evolutivos para encontrar configuraciones óptimas de pesos. Se empleará un enfoque de optimización multiobjetivo (e.g., NSGA-II, MOEA/D) [7] para equilibrar precisión y fairness.

4. Evaluación y validación.- Se medirán métricas de precisión como accuracy, F1-score. Se evaluará fairness mediante algunas métricas como Disparate Impact, Equalized Odds, Absolute Between-ROC Area (ABROCA) o las diferencias en tasas de falsos positivos/negativos entre grupos sensibles [8]. Se analizarán los trade-offs a través de la visualización de fronteras de Pareto. Se buscará la mejor configuración propuesta mediante técnicas como el *Compromise programming* [5]

5. Análisis de resultados .- Se compararán los resultados obtenidos con los distintos modelos y algoritmos evolutivos empleados. Se evaluará la capacidad del enfoque propuesto para mejorar la equidad sin una degradación significativa del rendimiento.

**Agradecimientos.** Trabajo parcialmente financiado por el Gobierno de Castilla-La Mancha, la Universidad de Castilla-La Mancha y los Fondos Europeos, UE, mediante los proyectos SBPLY/21/180225/000062 y 2022-GRIN-34437. Trabajo parcialmente financiado por MICIU/AEI/10.13039/501100011033 and ERDF, EU mediante el proyecto PID2022-139293NB-C32.

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
