# OpenReview forum: "Estudio sobre el ajuste óptimo de pesos en SPODEs para balancear Accuracy y Fairness en el clasificador probabilístico AODE"
_MAEB/2025/Projects_Track — MAEB 2025 Proyectos_

### Official Review · Reviewer_LXXw · 2025-03-07
**About Balancing Accuracy and Fairness for the AODE probabilistic classifier**

**Rating:** 5
**Confidence:** 3

**Review:**

I think the paper describes an interesting protocol to parameterize the Average One-Dependence Estimators using a weighting system for the SPODEs in order to balance the accuracy and fairness in the AODE probabilistic classifier. I consider the achievements reached in the paper are suitable to be presented at the MAEB 2025 conference. Concerning the paper format, it is a bit astonish that the paper written in Spanish is full  of acronyms and expressions in English. I think it would be much better to have written the paper in English.

---

### Official Review · Reviewer_TQqR · 2025-03-17
**Review for the project "Estudio sobre el ajuste óptimo de pesos en SPODEs para balancear Accuracy y Fairness en el clasificador probabilístico AODE"**

**Rating:** 4
**Confidence:** 4

**Review:**

This proposal on the optimal adjustment of weights in SPODEs to balance Accuracy and Fairness in AODE is interesting, especially because of its focus on fairness in probabilistic classifiers, a relevant topic in AI ethics. Most research on fairness in classification has focused on Naïve Bayes (NB), Random Forest, SVMs or neural networks, but little has been done with AODE. The proposal is innovative because it applies multi-objective optimization to AODE for fairness, but it remains to be seen whether this provides a clear value with respect to traditional methodologies. Other approaches, such as retraining with balanced data or fairness-based regularization, could be equally effective without such computationally expensive optimization.

The proposal assumes that varying the weights of SPODEs will help improve fairness without affecting accuracy too much, but this is not clear a priori. It could be that some SPODEs contain systematic biases, and simply adjusting their weights does not really eliminate the problem. Adjusting the weights of SPODEs may not actually eliminate the bias, but only redistribute it, since adjusting weights may cause certain relationships between sensitive attributes and the class to be minimized, but does not necessarily eliminate the source of the bias in the data. An alternative approach would be to identify and eliminate the problematic dependencies explicitly rather than simply reducing their influence with weights.

The methodology mentions metrics such as Statistical Parity, Disparate Impact, Differential Fairness, but it is not clear whether fairness at the individual level (e.g., similarity in predictions between similar individuals) will also be assessed. This can be a problem because adjusting weights can reduce biases between groups but create new internal discriminations.

The multi-objective optimization approach (NSGA-II, MOEA/D) is not new. The use of evolutionary algorithms to optimize weights in classifiers is a well-known technique. The NSGA-II and MOEA/D methods have already been applied to other fairness problems in AI (e.g., Blank & Deb 2020, and Boulitsakis-Logothetis 2022), but not directly to AODEs with weights in SPODEs. The proposal does not seem to have considered that these algorithms can be very expensive on large datasets. It is not mentioned whether more efficient techniques (such as gradient-based optimization) have been explored to make the adjustment faster. It is also not clear how the constraints will be set. For example, when adjusting the weights, will there be any constraints to prevent the accuracy from falling too much? In multi-objective optimization, it is common to define minimum thresholds (e.g. "accuracy ≥ 80%"), but the document does not specify this.

The use of well-known datasets (COMPAS, Adult Income, German Credit) is good for validating the approach, but it would be desirable to see what happens on non-academic data. If a fairness model cannot be easily applied to real systems (because it is too slow or difficult to calibrate), its practical relevance decreases. Benchmarks against existing methods should be included, such as retraining with fairness constraints or data reweighting to see if the weight adjustment is really better.

In summary, the proposal is good because the application of multi-objective optimization to fairness in AODE is little explored and could be an interesting contribution. The use of known datasets facilitates comparison with other methods and the analysis of Pareto fronts is a good way to understand the trade-offs between fairness and precision. However, the proposal has some weaknesses and shortcomings. For example, it is not clear or substantiated that weight adjustment is the best way to correct biases in AODE. In addition, the computational cost can be high, and other simpler methods could offer similar solutions with less computation. In my opinion, a clearer justification on what results are expected and how the acceptability thresholds will be set is missing. An explicit comparison with other approaches should be included for fairness. It is not said whether the proposal will be compared with methods such as Equalized Odds Post-Processing, Re-weighting according to class disparity, or Fair adversarial training. Without this comparison, it will not be possible to claim that the approach is better or more useful.

---

### Decision · Program_Chairs · 2025-03-20

Accept